# Serum Levels of Plasmalogens and Fatty Acid Metabolites Associate with Retinal Microangiopathy in Participants from the Finnish Diabetes Prevention Study

**DOI:** 10.3390/nu13124452

**Published:** 2021-12-14

**Authors:** Vanessa Derenji de Mello, Tuomas Selander, Jaana Lindström, Jaakko Tuomilehto, Matti Uusitupa, Kai Kaarniranta

**Affiliations:** 1Institute of Public Health and Clinical Nutrition, University of Eastern Finland, FI-70211 Kuopio, Finland; Matti.Uusitupa@uef.fi; 2Science Service Center, Kuopio University Hospital, FI-70029 Kuopio, Finland; Tuomas.Selander@kuh.fi; 3Public Health Prevention Unit, Finnish Institute for Health and Welfare, FI-00271 Helsinki, Finland; Jaana.Lindstrom@thl.fi (J.L.); Jaakko.Tuomilehto@thl.fi (J.T.); 4Department of Public Health, University of Helsinki, FI-00014 Helsinki, Finland; 5Diabetes Research Group, King Abdulaziz University, Jeddah 21589, Saudi Arabia; 6Department of Ophthalmology, Kuopio University Hospital, FI-70029 Kuopio, Finland; kai.kaarniranta@uef.fi; 7Institute of Clinical Medicine, Faculty of Health Sciences, University of Eastern Finland, FI-70211 Kuopio, Finland

**Keywords:** diabetic nephropathy, plasmalogens, serum fatty acids, lifestyles

## Abstract

Diabetic retinopathy (DR) is the most common microvascular complication of diabetes, and retinal microaneurysms (MA) are one of the first detected abnormalities associated with DR. We recently showed elevated serum triglyceride levels to be associated with the development of MA in the Finnish Diabetes Prevention Study (DPS). The purpose of this metabolomics study was to assess whether serum fatty acid (FA) composition, plasmalogens, and low-grade inflammation may enhance or decrease the risk of MA. Originally, the DPS included 522 individuals (mean 55 years old, range 40–64 years) with impaired glucose tolerance who were randomized into an intervention (*n* = 265) or control group (*n* = 257). The intervention lasted for a median of four years (active period), after which annual follow-up visits were conducted. At least five years after stopping the intervention phase of DPS, participants classified as MA negative (*n* = 115) or MA positive (*n* = 51) were included in the current study. All these participants were free of diabetes at baseline (WHO 1985) and had high-sensitive C-reactive protein (hs-CRP), serum FA composition, and selected lipid metabolites measured during the active study period. Among the markers associated with MA, the serum plasmalogen dm16:0 (*p* = 0.006), the saturated odd-chain FA 15.0 (pentadecanoic acid; *p* = 0.015), and omega-3 very long-chain FAs (*p* < 0.05) were associated with a decreased occurrence of MA. These associations were independent of study group and other risk factors. The association of high serum triglycerides with the MA occurrence was attenuated when these MA-associated serum lipid markers were considered. Our findings suggest that, in addition to *n*-3 FAs, odd-chain FA 15:0 and plasmalogen dm16:0 may contribute to a lower risk of MA in individuals with impaired glucose tolerance. These putative novel lipid biomarkers have an association with MA independently of triglyceride levels.

## 1. Introduction

Diabetic retinopathy (DR) is the most common microvascular complication of diabetes and the leading cause of blindness in working-aged people with diabetes [1,2]. Retinal microaneurysms are one of the first detected abnormalities in fundus examinations associated with DR. The long duration of diabetes, poor glycemic control, and presence of hypertension are the main risk factors of DR [3]. The clinical signs of DR are identical in both type 1 and type 2 diabetes mellitus (T2DM), which include the development of microaneurysms (MA) (focal outpouchings of the retinal vessels), telangiectatic vessels, hemorrhages, exudates, capillary closure, and retinal neovascularization [4]. Elevated serum cholesterol and triglyceride concentrations have been reported to be a risk factor for DR [5,6], and it has been suggested that permeability changes in the retinal microvasculature result in extravascular accumulations of lipoprotein deposits with a consequent loss of function in the surrounding retinal cells [7]. High serum cholesterol and triglyceride concentrations are also associated with leaking MAs in diabetic macular edema that may lead to permanent visual loss [8]. Complexes of lipoproteins and macrophages are visualized in fundus photography as hard exudates suggesting that increased serum lipid levels are a risk factor for this advanced form of exudates and other macular complications in diabetes [8,9].

During the early stages of diabetes, retinal changes are primarily monitored by regular fundus photograph examinations. The early recognition of diabetes and DR is important to prevent or slow down the progression of late complications [10]. We recently documented that a lifestyle intervention program to prevent T2DM decreased the risk of retinal MA in the intervention group of individuals with impaired glucose tolerance (IGT) at baseline [11]. In this same study, we reported that elevated serum triglycerides were associated with the development of early diabetic microangiopathy in the Finnish Diabetes Prevention Study (DPS) [11]. In the DPS, which recruited overweighed participants with IGT [12,13], a lower risk of developing T2DM was associated with better insulin sensitivity (IS) and preserved β-cell capacity, both of which were related to healthy lifestyles [14].

Among the factors underlying elevated serum triglycerides are lifestyle (diet and physical activity), obesity, and T2DM. It is well known that dietary factors, in particular omega-3 fatty acids (FAs), are involved in the metabolism of triglycerides, and that these FAs have also many other metabolic effects [15], which are probably more directly associated with the development of early microvascular abnormalities related to diabetes. Among these factors, plasmalogens, fatty acid composition of cell membranes, inflammation, and oxidative stress could also be involved in the pathophysiological processes leading to early DR [16,17,18,19,20,21].

The association between dyslipidemia and DR has been extensively investigated, but a conclusive link or causality remains elusive. In the present study, we investigated putative underlying lipid-related factors that could explain the association of serum triglycerides with the development of early diabetic microangiopathy in a sub-group of participants of the Finnish DPS [11]. Specifically, the purpose of this metabolomics study was to assess whether serum FA composition, plasmalogens, and low-grade inflammation may enhance or decrease the risk of early DR.

## 2. Materials and Methods

### 2.1. Participants and Study Design

A total of 522 individuals with IGT at baseline (between 1993 and 1998) participated in the multicenter DPS randomized controlled trial study and were randomly assigned to the intensive lifestyle intervention or the control group [22]. The main goals of the lifestyle intervention delivered by clinical nutritionists were a reduction in body weight of 5% or more, total fat and saturated intakes less than 30% and 10% of energy consumed, respectively, fiber intake more than 15 g/1000 kcal, and moderate physical activity of at least 30 min/day [23]. Detailed information of the study design and interventions has been presented previously [12,23]. All the participants were examined on a yearly basis for their fasting and 2 h glucose after an oral glucose tolerance test (OGTT), anthropometry, blood pressure, serum lipids, and several other parameters. The total time of the follow-up with intervention (median 4 years, range 1–6 years) and post-intervention period combined was up to 15 years [22], and 246 of the participants were diagnosed with T2DM during the total follow-up. Between the years 2002 and 2006 (at least five years after the intervention phase), the participants in four of the five study centers were invited to take part in the fundus photography examination (dilated pupils, one field 30-degree), as previously described, and this resulted in data being available for 201 participants [11]. Of those, 166 had serum lipid metabolomics measurements available for the present study. Therefore, participants (MA negative group: MA_neg, *n* = 115; MA positive group: MA_pos, *n* = 51) analyzed in this report include those who did not have diabetes according to the revised WHO 1999 criteria at baseline and had stored serum samples available from the active study period for serum FA composition and selected metabolites measurements (Table 1). According to what has been previously published, the intervention group had less frequent MAs (24%, *p* = 0.029) compared with the control group (38%) [11]. Otherwise, there were no significant differences in the incidence of early retinal changes identified from ocular photographs between the control and intervention groups, as in [11]. 

### 2.2. Biochemical Analyses

Plasma glucose and serum insulin levels were determined as previously described [14]. High-sensitive C-reactive protein (hs-CRP) was measured in fasting serum at a one-year follow-up examination and thereafter yearly during the mean four-year intervention (active study) period using an IMMULITE^®^ 2000 Systems Analyzer (Siemens Healthcare Diagnostics, Inc. Tarrytown, NY, USA) [24,25]. 

The total serum FA composition and fatty acid metabolites (plasmalogens) were measured by gas chromatography using stored −80 °C serum samples taken during the active intervention period, as previously described [26]. The proportions of each of the FAs are expressed as molar percentages. The plasmalogens covered and detected by the applied method are the plasmalogen palmitic (dm16:0) and plasmalogen stearic (dm18:0). The intra- and inter-assay CV % for individual FAs and FA metabolites were ≤10% and ≤12%, respectively. 

Stearoyl-coenzyme A desaturases-1 and -2 (SCD-1, SCD-2) and delta 5 and 6 desaturase enzymes (D5D and D6D), the latter of which are suggested to predict the incidence of T2DM and may be involved in the development of T2DM [27,28,29,30,31,32,33], were estimated as follows: SCD-1 and SCD-2 activities as the product-to-precursor ratio of 16:1*n*-7/16:0 and 18:1*n*-9/18:0, respectively; and D5D and D6D activities by product-to-precursor ratios of 20:4*n*-6/20:3*n*-6 and 18:3*n*-6/18:2*n*-6, respectively. 

### 2.3. Statistical Methods

Data for investigated variables from the first two years of the trial were averaged and these averaged data were expressed as means with standard deviations for both MA (yes = positive: pos/no = negative: neg) groups. Group difference in years 1 and 2 between these MA groups were compared by a linear mixed effect (LME) model for repeated measurements adjusted for study group. In full-adjusted LME analysis, we also included in the model BMI, HbA1c, age, sex, and either fasting triglycerides or hs-CRP levels. Beta coefficients and Z-scores with 95% confidence intervals (CI) were extracted from LME model for each marker to measure the adjusted group difference between MA groups during the first two years. Statistical analyses were executed by R statistical software, version 4.0.4 (R Foundation for Statistical Computing, Vienna, Austria). Two-sided *p*-values < 0.05 were set to indicate statistically significant results.

## 3. Results

### 3.1. Serum FAs and Plasmalogens Relate with the Occurrence of MA Independently of the Study Group

Among the FAs and their metabolites analyzed and illustrated in Table 2, we first applied simple statistical models, adjusting only for the study group. While the serum proportion of the odd-chain saturated FA 15:0 (pentadecanoic acid; *p* = 0.015) was inversely associated with MA, the monounsaturated FAs (MUFAs) 16:1*n*-7 (palmitoleic acid; *p* = 0.026), 18:1*n*-9 (oleic acid; *p* < 0.001), and 20:3*n*-9 (mead acid; *p* = 0.013) were directly associated with the occurrence of MA (Table 2). 

Among the polyunsaturated FAs, 20:2*n*-6 (eicosadienoic acid; *p* = 0.01) was higher in the MA_pos group while all 20:5*n*-3 (eicosapentaenoic acid; *p* = 0.030), 22:5*n*-3 (docosapentaenoic acid; *p* = 0.018), and 22:6*n*-3 (docosahexaenoic acid; *p* = 0.002) were lower in the MA_pos group (Table 2). D5D (*p* = 0.021), but not D6D (*p* = 0.21), which was inversely associated with occurrence of MA (Table 2).

Even though only odd-chain 15:0 out of the saturated FA proportions associated with the occurrence of MA, the SCD-1 estimate (*p* = 0.032) was also significantly but directly associated with the presence of MA, but not SCD-2 (*p* = 0.806) (Table 2). Of note, serum plasmalogens dm16:0 (*p* = 0.006) and dm18:0 (*p* = 0.044) were both inversely associated with the occurrence of MA (Table 2). 

In the full-adjusted models (Figure 1), the associations remained unchanged for the FAs, estimated desaturases and dm16:0, except for both SCD-1 (*p* = 0.074) and dm18:0 (*p* = 0.103), which lost their significance in relation to MA occurrence.

### 3.2. MA-Associated Metabolites Attenuate the Relationship of Serum Triglycerides with MA Occurrence

Applying full-adjusted models, the previously observed direct association of serum triglycerides levels with MA, suggesting the harmful effect on the development of MA [11], was either attenuated or completely lost when adjusted for the FAs, plasmalogens, or calculated desaturase activities that we found to be associated with the presence of MA (Table 3).

### 3.3. Low-Grade Inflammation Is Not Associated with the Occurrence of MA

We did not find any significant association of hsCRP with MA in either of these models (simple, *p* = 0.87; full-adjusted, *p* = 0.704). The associations found for FA proportions, plasmalogens, or desaturases with the presence of MA were independent of hsCRP (Table 4).

## 4. Discussion

We found that serum plasmalogens, particularly the one with palmitic acid in its structure, together with saturated odd-chain FA 15.0 (pentadecanoic acid) and omega-3 very long-chain FAs were associated with decreased occurrence of MA. Conversely, MUFAs were associated with an increased the occurrence of MA. 

Plasmalogens represent an important class of phospholipids whose presence is required for proper brain and eye development [34]. In adults at high risk of developing T2DM, the plasmalogens containing palmitic acid in their structure were found to be increased as a result of an 18–24-week Nordic healthy diet. This Nordic diet was rich in whole grains, fruits, vegetables, berries, vegetable fat, and fish, along with low-fat milk products and low-fat meat choices [35]. Other plasmalogens species are also among the serum metabolites associated with higher nut intake in adults without T2DM [36]. Therefore, increased levels of these lipid metabolites in participants who did not have MA could be partly associated with the protective effects of a healthy diet. Plasmalogens form a specific subclass of glycerophospholipids characterized by the presence of a vinyl-ether bond at the sn-1 position of the glycerol backbone, instead of an ester bond, as seen in diacyl-glycerophospholipids. Plasmalogens have antioxidative potential [37] and increased lipid oxidation is linked to decreased plasmalogen concentrations [18]. Lipid oxidation is in turn related to the development and progression of DR [38,39]. In the eye, plasmalogens account for 30% of glycerolphosphatidyletanolamines in retinal pigment epithelial cells that are highly exposed to oxidative damage. In these cells, plasmalogens have been suggested to play protective roles against oxidative stress [34]. 

We observed that the odd-chain FA 15:0 (pentadecanoic acid) and omega-3 very long-chain FAs were associated with a decreased occurrence of MA (Figure 2). Oxidative stress and inflammation have been proven to be critical contributors to the development of DR [39,40]. It has been reported that in vivo oxidative stress measured in plasma and urine from patients with T2DM was reduced by the supplementation of *n*-3 FAs [41]. Overall, data coming from human and animal studies suggest *n*-3 FAs in the prevention and treatment of retinal diseases due to their effects on improving retinal oxidative stress and inflammation [42]. Although we were not able to directly link lower systemic inflammation with less occurrence of MA, we cannot rule out an anti-inflammatory effect on the retina of higher *n*-3 and 15:0 FAs. In some studies, a beneficial effect of sources of *n*-3 FAs on risk markers of T2DM [43,44] and a possible protective effect of these FAs on the prevalence and progression of diabetic MA have also been shown [45]. We have previously found that MA occurrence was lower in a subpopulation of the DPS allocated to the lifestyle intervention group [11]. Therefore, the relationship of lifestyle changes with healthier eyes in the long term could be explained by dietary changes that are involved in protective biochemical pathways that may prevent later DR development. *n*-3 long-chain FAs also predicted a decreased risk of T2DM in the DPS [26] and this was confirmed in a recent meta-analysis [46]. The odd chain 15:0 FA may be derived from milk fat, but also, e.g., from fish, whereas the role of gut microbiota after dietary interventions rich in fiber intake in its formation is not yet fully understood [47,48,49,50]. Nevertheless, high fiber intake has been associated with a lower risk of developing T2DM [49,51], which may be associated with other healthy aspects of a higher fiber intake.

The increased occurrence of MA with MUFAs might be attributed to its possible pro-angiogenic role of in retina [52,53]. Nevertheless, inverse associations have been observed between total MUFAs and oleic acid intakes with DR, in persons with T2DM [54]. Our results related to MUFA in blood do not necessarily reflect their dietary intake, but rather metabolic disturbances, e.g., as in [55,56]. In the typical Finnish diet, the main source of oleic acid is milk fat that is rich in oleic acid (about 25%) [57].

Strengths of the present study include the well-characterized and homogenous study population comprising overweight and obese people with IGT. The outcome, MA, was determined without information concerning any metabolic parameters. The major limitation in our study is that we do not have baseline data on retinal changes. We also did not perform fundus photography examination in all of the DPS participants. However, the overall results retrieved from this study are plausible and thus add to the scientific field of pathophysiology and prevention of DR in T2DM. 

Our findings support the view that dyslipidemia and specific lipid metabolites, in addition to high serum triglycerides, play an important pathogenic role in the development of DR. The benefit of *n*-3-long chain FAs and C15:0 for eye health has also been related to a decreased risk of T2DM.

## 5. Conclusions

In conclusion, here, we show putative novel lipid biomarkers beyond serum triglycerides that are associated with MA. Our new findings suggest that plasmalogen dm16:0 may contribute to a lower risk of MA in overweight and obese individuals with IGT, especially in those who have elevated serum triglycerides. Our observation may open new lines of lipid-lowering therapies in future investigations aiming at preventing retinal complications in persons with hyperglycemia. In Figure 2, we propose to include the present findings among the factors beyond dyslipidemia that may enhance or decrease the risk of early nephropathy.

## Figures and Tables

**Figure 1 nutrients-13-04452-f001:**
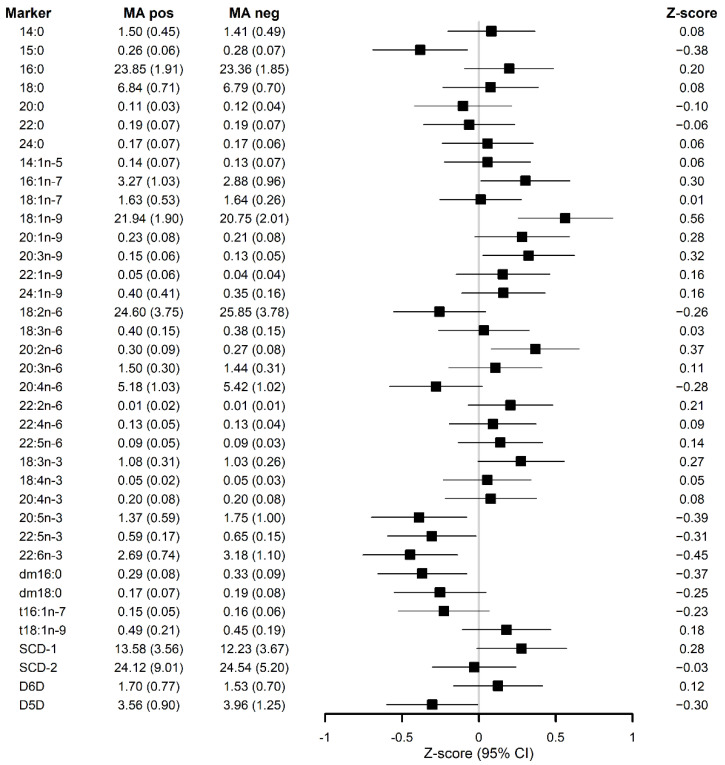
Z-scores for serum lipid metabolite markers and their standard deviations (SD) in the forest plot describing the association of each serum lipid metabolite markers with the occurrence of microaneurysms after applying full-adjusted linear mixed effect (LME) model for repeated measurements adjusted for study group, BMI, HbA1c, age, sex and fasting serum triglycerides. dm: plasmalogen; t: trans; SCD-1: Stearoyl-coenzyme A desaturase 1; SCD-2: Stearoyl-coenzyme A desaturase 2; D5D: delta 5 desaturase enzyme; D6D: delta 5 desaturase enzyme.

**Figure 2 nutrients-13-04452-f002:**
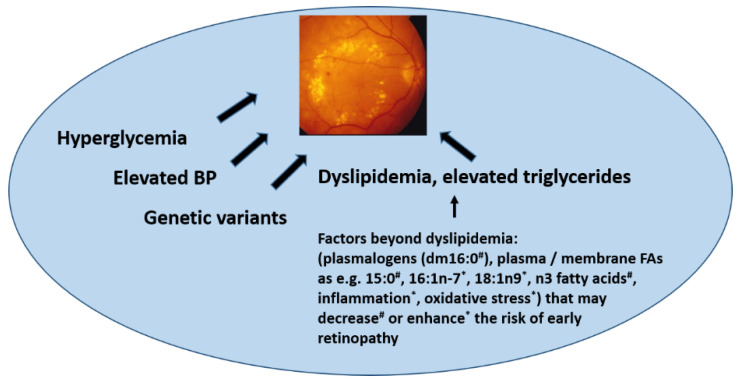
Factors associated with the development of retinopathy. BP: blood pressure FA: fatty acids. With symbols are the proposed factors we find associated with retinal microaneurysms occurrence in the present study (# positively associated; * negatively associated).

**Table 1 nutrients-13-04452-t001:** Characteristics of the participants.

	MA Neg	MA Pos	*p*-Value
*n*	115	51	
Study group, *n* (I/C)	67/48	21/30	0.062
Sex, *n* (men/women)	35/80	9/42	0.126
Age, years	54.4 ± 7.5	52.0 ± 6.2	0.036
BMI, kg/m^2^	31.3 ± 5.0	31.7 ± 5.1	0.641
Triglycerides, mmol/L	1.60 ± 0.68	1.85 ± 0.68	0.035
HbA1c, %	5.53 ± 0.56	5.35 ± 0.55	0.051
Hs-CRP, mg/L	0.74 ± 0.99	0.68 ± 1.01	0.739

MA: microangiopathy neg: negative pos: positive. I: intervention. C: control. Hs-CRP: high-sensitive C-reactive protein.

**Table 2 nutrients-13-04452-t002:** Serum fatty acids, plasmalogens metabolites and estimated desaturases according to the occurrence of retinal microangiopathy (MA), and associations of MA with the serum fatty acids, plasmalogens metabolites and estimated desaturases.

Serum Lipid Marker	MA Pos ^1^	MA Neg ^1^			
mmol%	Mean ± SD	Mean ± SD	β Value	CI	*p* ^2^
14:0 myristic	1.5 ± 0.45	1.41 ± 0.49	0.04	−0.10; 0.18	0.58
15:0 pentadecanoic	0.26 ± 0.06	0.28 ± 0.07	−0.03	−0.05; −0.01	0.015
16:0 palmitic	23.85 ± 1.91	23.36 ± 1.85	0.34	−0.23; 0.91	0.25
18:0 stearic	6.84 ± 0.71	6.79 ± 0.7	0.06	−0.15; 0.27	0.58
20:0 arachidic	0.11 ± 0.03	0.12 ± 0.04	0.00	−0.02; 0.01	0.50
22:0 behenic	0.19 ± 0.07	0.19 ± 0.07	0.00	−0.02; 0.02	0.92
24:0 lignoceric	0.17 ± 0.07	0.17 ± 0.06	0.00	−0.02; 0.02	0.69
14:1*n*-5 myristoleate	0.14 ± 0.07	0.13 ± 0.07	0.01	−0.01; 0.03	0.56
16:1*n*-7 palmitoleic	3.27 ± 1.03	2.88 ± 0.96	0.35	0.04; 0.66	0.026
18:1*n*-7 vaccenic	1.63 ± 0.53	1.64 ± 0.26	0.00	−0.10; 0.10	0.95
18:1*n*-9 oleic	21.94 ± 1.9	20.75 ± 2.01	1.16	0.54; 1.78	<0.001
20:1*n*-9 gondoic	0.23 ± 0.08	0.21 ± 0.08	0.02	0.00; 0.04	0.11
20:3*n*-9 mead	0.15 ± 0.06	0.13 ± 0.05	0.02	0.00; 0.04	0.013
22:1*n*-9 erucic	0.05 ± 0.06	0.04 ± 0.04	0.01	−0.01; 0.02	0.41
24:1*n*-9 nervonic	0.4 ± 0.41	0.35 ± 0.16	0.05	−0.02; 0.12	0.19
18:2*n*-6 linoleic	24.6 ± 3.75	25.85 ± 3.78	−0.95	−2.12; 0.21	0.11
18:3*n*-6 γ-linolenic	0.4 ± 0.15	0.38 ± 0.15	0.02	−0.03; 0.06	0.48
20:2*n*-6 eicosadienoic	0.3 ± 0.09	0.27 ± 0.08	0.03	0.01; 0.05	0.01
20:3*n*-6 dihomo-γ-linolenic	1.5 ± 0.3	1.44 ± 0.31	0.06	−0.03; 0.16	0.20
20:4*n*-6 arachidonic	5.18 ± 1.03	5.42 ± 1.02	−0.24	−0.55; 0.07	0.13
22:2*n*-6 docosadienoic	0.01 ± 0.02	0.01 ± 0.01	0.00	0.00; 0.01	0.12
22:4*n*-6 adrenic	0.13 ± 0.05	0.13 ± 0.04	0.01	−0.01; 0.02	0.30
22:5*n*-6 osbond	0.09 ± 0.05	0.09 ± 0.03	0.01	−0.01; 0.02	0.31
18:3*n*-3 ꭤ-linolenic	1.08 ± 0.31	1.03 ± 0.26	0.05	−0.03; 0.13	0.20
18:4*n*-3 stearidonic	0.05 ± 0.02	0.05 ± 0.03	0.00	−0.01; 0.01	0.68
20:4*n*-3 eicosatetraenoic	0.2 ± 0.08	0.2 ± 0.08	0.00	−0.02; 0.03	0.77
20:5*n*-3 eicosapentaenoic	1.37 ± 0.59	1.75 ± 1	−0.42	−0.70; −0.14	0.003
22:5*n*-3 docosapentaenoic	0.59 ± 0.17	0.65 ± 0.15	−0.06	−0.10; −0.01	0.018
22:6*n*-3 docosahexaenoic	2.69 ± 0.74	3.18 ± 1.1	−0.50	−0.82; −0.19	0.002
dm16:0	0.29 ± 0.08	0.33 ± 0.09	−0.04	−0.06; −0.01	0.006
dm18:0	0.17 ± 0.07	0.19 ± 0.08	−0.02	−0.05; 0.00	0.044
t16:1*n*-7	0.15 ± 0.05	0.16 ± 0.06	−0.01	−0.03; 0.00	0.13
t18:1*n*-9	0.49 ± 0.21	0.45 ± 0.19	0.04	−0.02; 0.10	0.17
SCD-1 ^3^	13.58 ± 3.56	12.23 ± 3.67	1.26	0.11; 2.41	0.032
SCD-2 ^3^	24.12 ± 9.01	24.54 ± 5.2	−0.23	−2.02; 1.57	0.81
D5D ^3^	3.56 ± 0.90	3.96 ± 1.25	−0.42	−0.77; −0.07	0.021
D6D ^3^	1.7 ± 0.77	1.53 ± 0.7	0.14	−0.08; 0.35	0.21

^1^ mean values sampled at year 1 and 2 of the active trial period. ^2^ linear mixed effect (LME) model for repeated measurements adjusted for study group. dm: plasmalogen. t: trans. ^3^ expressed as ratios. SCD-1: Stearoyl-coenzyme A desaturase 1. SCD-2: Stearoyl-coenzyme A desaturase 2. D5D: delta 5 desaturase enzyme. D6D: delta 5 desaturase enzyme.

**Table 3 nutrients-13-04452-t003:** Association strength (β and CI) of serum triglycerides levels with MA, when MA-associated lipid metabolite markers were included in the model.

Serum Lipid Marker Included in the Model ^1^	Triglyceride’s Resulting β Value	CI	*p* ^1^
15:0	0.16	(−0.02; 0.34)	0.09
16:1*n*-7	0.10	(−0.07; 0.27)	0.24
18:1*n*-9	0.03	(−0.12; 0.18)	0.68
20:5*n*-3	0.13	(−0.04; 0.31)	0.14
22:5*n*-3	0.16	(−0.02; 0.34)	0.09
22:6*n*-3	0.13	(−0.04; 0.31)	0.14
dm16:0	0.06	(−0.09; 0.21)	0.41

^1^ each marker was included one by one as a covariate in the linear mixed effect (LME) model for repeated measurements in addition to study group, BMI, HbA1c, age, sex and serum fasting triglycerides. dm: plasmalogen.

**Table 4 nutrients-13-04452-t004:** Association of serum fatty acids, plasmalogens metabolites and estimated desaturases with the presence of retinal microangiopathy (MA) in full-adjusted models.

Serum Lipid Marker	β	CI	*p* ^1^
14:0 myristic	0.04	−0.10; 0.17	0.573
15:0 pentadecanoic	−0.03	−0.05; 0.00	0.016
16:0 palmitic	0.37	−0.17; 0.92	0.184
18:0 stearic	0.05	−0.16; 0.27	0.630
20:0 arachidic	0.00	−0.02; 0.01	0.521
22:0 behenic	0.00	−0.03; 0.02	0.675
24:0 lignoceric	0.00	−0.02; 0.02	0.705
14:1*n*-5 myristoleate	0.00	−0.02; 0.02	0.698
16:1*n*-7 palmitoleic	0.30	0.01; 0.59	0.042
18:1*n*-7 vaccenic	0.00	−0.09; 0.10	0.930
18:1*n*-9 oleic	1.16	0.53; 1.79	<0.001
20:1*n*-9 gondoic	0.02	0.00; 0.05	0.075
20:3*n*-9 mead	0.02	0.00; 0.03	0.035
22:1*n*-9 erucic	0.01	−0.01; 0.02	0.317
24:1*n*-9 nervonic	0.04	−0.03; 0.12	0.250
18:2*n*-6 linoleic	−0.98	−2.12; 0.16	0.094
18:3*n*-6 γ-linolenic	0.00	−0.04; 0.05	0.827
20:2*n*-6 eicosadienoic	0.03	0.01; 0.06	0.012
20:3*n*-6 dihomo-γ-linolenic	0.03	−0.06; 0.13	0.485
20:4*n*-6 arachidonic	−0.29	−0.60; 0.02	0.070
22:2*n*-6 docosadienoic	0.00	0.00; 0.01	0.147
22:4*n*-6 adrenic	0.00	−0.01; 0.02	0.529
22:5*n*-6 osbond	0.01	−0.01; 0.02	0.321
18:3*n*-3 ꭤ-linolenic	0.08	0.00; 0.15	0.056
18:4*n*-3 stearidonic	0.00	−0.01; 0.01	0.709
20:4*n*-3 eicosatetraenoic	0.01	−0.02; 0.03	0.608
20:5*n*-3 eicosapentaenoic	−0.35	−0.63; −0.07	0.015
22:5*n*-3 docosapentaenoic	−0.05	−0.09; 0.00	0.039
22:6*n*-3 docosahexaenoic	−0.46	−0.77; −0.14	0.005
dm16:0	−0.03	−0.06; −0.01	0.013
dm18:0	−0.02	−0.04; 0.00	0.099
t16:1*n*-7	−0.01	−0.03; 0.00	0.132
t18:1*n*-9	0.04	−0.02; 0.09	0.223
SCD-1	1.02	−0.05; 2.10	0.064
SCD-2	−0.21	−2.04; 1.63	0.825
D5D	−0.35	−0.70; −0.01	0.046
D6D	0.09	−0.12; 0.30	0.400

^1^ each marker was included one by one as a covariate in the linear mixed effect (LME) model for repeated measurements in addition to study group, BMI, HbA1c, age, sex and serum high-sensitive C-reactive protein. SCD-1: Stearoyl-coenzyme A desaturase 1. SCD-2: Stearoyl-coenzyme A desaturase 2. D5D: delta 5 desaturase enzyme. D6D: delta 5 desaturase enzyme.

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
