# Peer review of "Serum Levels of Plasmalogens and Fatty Acid Metabolites Associate with Retinal Microangiopathy in Participants from the Finnish Diabetes Prevention Study"

_nutrients, 2021, doi:10.3390/nu13124452_

Round 1

Reviewer 1 Report

This article builds on the authors’ previously published study, demonstrating that elevated triglyceride levels are associated with retinal microaneuysms in diabetes for 522 individuals with impaired glucose tolerance. The current study assesses the metabolomics of 166 participants (51 of which had detectable retina microaneurysms) from the previous study at least five years after a lifestyle intervention of Nordic “healthy diet” delivered by clinical nutritionists. The research found that plasmalogens were associated with a decrease in microaneurysms, whilst monosaturated fatty acids were associated with an increase in the occurrence of retinal microaneurysms. This study reinforces the theory that dyslipidemia plays a role in the development of diabetic retinopathy. It provides additional elements to the field, notably that some specific fatty acids may provide benefits to eye health. I recommend this study for publication with minor changes.

Comments

  • Figure 1 does not add much to the introduction. However, adding some details of the study’s finding to the diagram would make a compelling figure for the discussion. I suggest including which biomarkers are beneficial, which aren’t and how they contribute to the progression of diabetic retinopathy.

Minor Comments

  • Grammar needs to be corrected throughout the manuscript, and in particular the first sentence of the introduction
  • I recommend adding a little bit more background on the biochemistry of plasmalogens

Author Response

Figure 1 does not add much to the introduction. However, adding some details of the study's finding to the diagram would make a compelling figure for the discussion. I suggest including which biomarkers are beneficial, which aren't and how they contribute to the progression of diabetic retinopathy.

Answer: As suggested by this reviewer, we switched Figure 1 (now Figure 2) to the discussion section. We introduced the main biomarkers we found to be related to DR and add their direction.

Minor Comments

Grammar needs to be corrected throughout the manuscript, and in particular the first sentence of the introduction
I recommend adding a little bit more background on the biochemistry of plasmalogens

Answer: we revised the language of the manuscript text including the first sentence of the introduction section. We have described the background on the biochemistry of plasmalogens in the discussion rather than in the introduction, though, as follows: “Plasmalogens form a specific subclass of glycerophospholipids characterized by the presence of a vinyl-ether bond at the sn-1 position of the glycerol backbone, instead of an ester bond as seen in diacyl-glycerophospholipids.”.

Reviewer 2 Report

Reviewer comments and suggestions

The purpose of this study was to assess whether serum fatty acid (FA) composition, plasmalogens and low-grade inflammation could aggravate or decrease the risk of microaneurysms (MA). 

The data was taken from, the Diabetes Prevention Study (DPS) included 522 individuals (mean 55 years old, range 40-64 years) with impaired glucose tolerance who were randomized into an intervention (n=265) or control group (n=257). 

At least five years after stopping the intervention phase of DPS, participants classified as MA negative, n=115 or MA positive, n=51 were included in the current study. 

The study reported that participants were free of diabetes at baseline and had high-sensitive C-reactive protein (hs-CRP), serum FA composition, and selected lipid metabolites measured during the active study period. 

The study suggests that n-3 FAs, odd-chain FA 15:0 and plasmalogen dm16:0 may contribute to a lower risk of MA in individuals with impaired glucose tolerance. 

Decision: Minor comments

Below are the comments for this paper to be incorporated in the revised version of the manuscript. 

  1. Line 53-54, need a reference
  2. Line 61-65, 210-213, 237-241, please avoid long sentences in the MS
  3. Table 2 and 4 (14, 15, 16, 18) better to name them here (in table)
  4. Line 226-227 better to name here table or figure
  5. Line 254-255 The authors can point one other limitation in the study as I understood the authors skip a few limitations
  6. Please explain this line with the line discussed in 260-262 at plasmalogen dm16:0 may contribute to a lower risk of MA in individuals with IGT (present in the conclusion)
  7. please check all the references, journal format it was not according to MDPI journals

Author Response

Line 53-54 need a reference.

Answer: We added the proper ref n. 8 earlier in the text rather than just at the end of the paragraph, as it was in the previous version of the manuscript.

Line 61-65, 210-213, 237-241, please avoid long sentences in the MS.

Answer: We made appropriated changes in the manuscript whenever possible throughout the text to avoid long sentences.

Table 2 and 4 (14, 15, 16, 18) better to name them here (in table). Line 226-227 better to name here table or figure.

Answer: We did not quite understand what the reviewer commented here. We though inserted a reference for Figure 2 in lines 226-227. We also included the fatty acids names in Tables 2 and 4.

Line 254-255 The authors can point one other limitation in the study as I understood the authors skip a few limitations

Answer: We have added as suggested another limitation of the study.

Please explain this line with the line discussed in 260-262 at plasmalogen dm16:0 may contribute to a lower risk of MA in individuals with IGT (present in the conclusion)

Answer: We added overweight and obese individuals so that it would be equal as what was described in lines 254-255

Please check all the references, journal format it was not according to MDPI journals

Answer: The references were reedited according to the Nutrients style editor using RefWorks program.